# Management Strategies for Napier Grass (*Pennisetum purpureum* Schumach cv Pakchong): Impact on Dry Matter Yield, Nutritive Characteristics and Cattle Growth

**DOI:** 10.3390/ani15091235

**Published:** 2025-04-27

**Authors:** Anamika Roy, Biplob Kumer Roy, Cameron Edward Fisher Clark, Muhammad Khairul Bashar, Nathu Ram Sarker, Nasrin Sultana, Md. Mostain Billah, Mohammad Al-Mamun, Mohammad Rafiqul Islam

**Affiliations:** 1Department of Animal Nutrition, Bangladesh Agricultural University, Mymensingh 2202, Bangladesh; royanamikadls30@gmail.com; 2Bangladesh Livestock Research Institute, Dhaka 1341, Bangladesh; biplobkumerroy@gmail.com (B.K.R.); nassul2003@yahoo.com (N.S.); mostainbau@gmail.com (M.M.B.); 3Gulbali Institute, Charles Sturt University, Wagga Wagga, NSW 2678, Australia; camclark@csu.edu.au; 4Krishi Gobeshona Foundation, Bangladesh Agricultural Research Council Complex, Farmgate, Dhaka 1215, Bangladesh; sarkernr62@yahoo.com; 5Dairy Science Group, Livestock Production and Welfare Group, Faculty of Science, The University of Sydney, Camden, NSW 2570, Australia; md.islam@sydney.edu.au

**Keywords:** Napier grass, leaf stage, biomass yield, nutritive value, in vitro, growth

## Abstract

Napier grass is widely used across the tropics and subtropics as a feed for dairy and beef cattle, but its quality is poor under current management with resultant poor animal productivity. Harvesting Napier grass at leaf stage six appears appropriate for feed nutritive value and cattle growth. Solutions are required to ameliorate yield loss associated with this management strategy, such as increasing plant density, and this should be the focus of further work.

## 1. Introduction

Napier grass (*Pennisetum purpureum* Schumach) is a key forage for tropical and subtropical smallholder animal production systems due to its robust growth [1,2,3]. Despite being a highly suitable forage species for these regions [4], Napier grass is deemed to be of low quality due to its high fiber content and low protein (70–100 g/kg DM) and energy (6.0–8.0 MJ/kg DM) levels [5,6], limiting the growth of animals. Farmers typically harvest Napier grass at a height of 150–200 cm or greater [7,8] in order to maximize forage production from their limited landholdings. Nevertheless, the majority of children in the same regions where Napier grass is grown are stunted and malnourished in part due to the low productivity of livestock and associated protein supply. Therefore, the challenge is to optimize the nutritional content of Napier grass while maintaining high biomass yields, with the aim of enhancing milk and meat production in tropical and subtropical regions. This, in turn, can contribute to the overall goal of ensuring food and nutrition security, as well as reducing malnutrition, stunting and wasting among children in these areas [9].

Recently, many developing countries in the tropics and subtropics are rapidly moving towards commercialization [10]. As incomes increase in these regions, more livestock products are demanded. However, the traditional management system of Napier grass is not a possible way to meet livestock demands. Evidence suggests that Napier grass can be managed to contain higher protein and energy to meet the nutritional requirements of ruminants for milk and meat production. This management follows the principles of leaf stage (LS), defoliation height (DH), severity height (SH) or defoliation/harvest intervals (DI) used to manage quality of grasses such as kikuyu (*Pennisetum clandestinum* ex chiov) or ryegrass. For example, Fulkerson et al. [11] demonstrated that kikuyu grass harvested at 4.5 LS (16 days DI) contains approximately 200 g CP/kg DM and 10 MJ ME/kg DM, and fed as fresh grass, supporting 15 L milk/cow/day in mid-lactating Holstein cows. Similarly, Napier grass is also shown to contain low fiber (300–350 g ADF/kg DM, 490–530 g NDF/kg DM) and high CP (170–250 g CP/kg DM) when harvested at a DI interval of 14–49 days [12]. Sileshi et al. [13] also showed that Napier grass harvested at a DI of 14–28 days contains 190–230 g CP/kg DM, 490–500 g NDF/kg DM and 11.7 MJ ME/kg DM. These findings suggest that high-quality Napier grass can be managed and produced by simple LS and defoliation management. However, one of the limitations is that none of these Napier grass-related experiments investigated year-round biomass yield and quality to determine the trade-off between yield and quality. Furthermore, there is no animal experimentation in the literature investigating the performance of animals with such high-quality Napier grass.

Here, we explored strategies for enhancing the yield and nutritional quality of Napier grass to boost milk and meat production in tropical and subtropical regions. The focus was on developing a best management practice (BMP) for this purpose. We hypothesized that a simple management practice of harvesting Napier grass at an earlier LS will increase nutritional composition by increasing the proportion of leaf to stem (Experiment 1) which, in turn, will increase cattle growth (Experiment 2).

## 2. Materials and Methods

### 2.1. Experiments, Location and Duration of the Study

Both experiments were conducted from May 2019 to April 2020 at the Pachutia Research Farm, Bangladesh Livestock Research Institute (BLRI), Dhaka, Bangladesh. The experimental site is known as Madhupur Tract, which is a part of Agro-ecological Zone (AEZ-28) of Bangladesh. It is located at an altitude of 4 m above sea level, with coordinates of 24°42′0″ N latitude, and 90°22′30″ E longitude. The soil of the site is clay textured, strongly acidic (pH 4.5–5.7) soil and contains low (<1.5%) organic matter. The climate of the site is characterized by a maximum temperature during April-July (summer months) accompanied by high rainfall (370 mm/month). Minimum temperature occurs during December-January (winter months) when it may drop below 10 °C accompanied by low rainfall (1–6 mm/month). Average humidity is 70.4% and the total annual rainfall is 1990 mm (https://weatherandclimate.com/bangladesh/dhaka/savar; accessed 20 April 2025).

Experiment 1 focused on Napier grass’s (cv Pakchong) nutritive value, which was conducted over 12 months. Experiment 2 aimed at cattle growth offering extreme treatments from Napier Pakchong grass, which was conducted over a period of 113 days. Napier Pakchong is a non-GMO hybrid, developed in Thailand by crossing *Pennisetum purpureum* and *Pennisetum glaucum* known as Pearl Millet Napier. It is commonly known as super Napier grass characterised by high yield with high nutritive value (https://agriculture.com.ph/2018/02/28/super-napier-planting-guide/; accessed 20 April 2025).

### 2.2. Experiment 1: Management of Napier Grass

#### Preparation of Experimental Plots and Management

Prior to start of the experiment, soil samples were collected from three different locations across the site at a depth of 0–30 cm. These samples were mixed thoroughly to make a composite sample and five composite soil sub-samples, each weighing approximately 100 g and prepared and sent to the Soil Resource Development Institute, Farmgate, Dhaka-1215 for analysis (Table 1). The soil type was silt loam (Sandy soil—23%, clay—13% and alluvial soil—64%).

The experimental land was then ploughed four times with a tractor, with weeds then removed manually from the ploughed land and finally, the land was levelled to ensure uniformity. A basal dose of urea (as N), triple superphosphate (as P), muriate of potash (as K) and gypsum at 297, 214, 56 and 90 kg/ha were applied according to the soil nutrient analysis report (Table 1). In addition, limestone was applied at a rate of 1976 kg/ha, 14 days before planting to adjust soil pH. Pakchong grass was planted by stem cuttings, where two cuttings were planted in a hole at a 45° angle and each cutting had two nodes. Plant-to-plant distance was 50 cm and row-to-row distance was also maintained at 50 cm. During the experimental period, urea fertiliser was applied to each plot at a rate of 249 g/cut (115 g N; Urea contains 46% N). All plots were irrigated to ensure non-limiting soil moisture for plant growth.

### 2.3. Experimental Design

The experimental design was a Randomized Complete Block Design (RCBD). Treatments included 3 LS (6, 9 and 14, respectively), each harvested at 3 SH (5, 10 and 20 cm above the ground level). These treatments were arranged in a 3 × 3 factorial arrangement and each treatment was replicated three times randomly across blocks, making a total of 27 plots. Each plot was 4 × 4 m^2^ and each plot consisted of 64 plants. Each plot was separated by a buffer alley of 2 m.

Plants after the initial planting reached 6, 9 and 14 LS on 28 June, 11 July and 18 August 2019, respectively, and were harvested (i.e., first harvest) on these dates for each SH of 5, 10 and 20 cm from the ground level. Subsequently, plants in each treatment were harvested at regular intervals throughout the year as plants reached their respective LS. During the harvesting time, we recorded morphological characteristics such as total leaf number (LS), normal and raised plant height (cm), the number of dead and green leaves, the number of nodes, the leaf–stem ratio and the biomass yield. Normal height is defined as the plant height at canopy and raised plant height was measured by straitening the canopy. Fresh biomass yield from each plot was weighed at the morning (from 8:00 to 9:00 O’clock) using a scale immediately after harvest and was converted to DM yield/plot according to the equation DM yield/plot = weight of fresh biomass × (%) DM.

To determine DM content, a representative sample (approximately 750 g) was taken and chopped to a length of 1–2 cm using Wiley Mill (Dietz-Motoren, KG, Elektromotorenfabrik, Dettinger-Tech, Germany). The chopped sample was mixed properly and a ~250 g sample was replicated twice and placed in paper bags. Sample bags were dried at 60 °C for 48 h in an oven [14]. Fresh (WW) and dried (DW) weight were recorded and DM content was calculated using the following formula:

% DM = DW/WW × 100 Here, WW; total sample weight while wet, DM; total sample weight while dry.

### 2.4. Napier Grass Nutritive Value

All grass samples were ground up using a 1.0 mm mesh sieve (Retsch GmbH, Haan, Germany) for chemical analysis; adhering to the procedure established by the Association of Official Analytical Chemists (AOAC) [15] for the determination of dry matter (DM), crude protein (CP), organic matter (OM) and ash. The ADF and NDF were quantified according to the methodology established by Van Soest et al. [16]. All samples were analyzed at the Animal Nutrition Laboratory, BLRI, Savar, Dhaka-1341 with duplicates, and mean values were recorded.

A bomb calorimeter (C 5000; IKa-Werke GmbH and Co. KG, Staufen, Germany) was used to assess the gross energy (GE) applying benzoic acid as a standard.

Oxalate content was determined using the method described by Rahman et al. [17] through HPLC and WSC by Smith [18].

In vitro digestibility was determined, as described by Asanuma et al. [19]. Fresh rumen fluid was collected from two ruminally fistulated RCC (Red Chittagong Cattle) bulls fed ad libitum Napier grass containing 12% CP and supplied 1.5% body weight of the concentrate mixture, which consisted of 33% wheat bran, 20% khesari bran, 15% rice bran, 10% crushed maize, 10% crushed wheat, 5% soybean, 2% vitamin, mineral premix, 1% DCP, 1% salt and 3% molasses. Drinking water supplied to the bulls ad libitum.

In brief, approximately 0.5 g of DM of feed samples was weighed and placed into 100 mL serum bottles, ensuring that the sample did not stick to the wall of the bottle. The serum bottle was sealed with a rubber septum stopper and aluminum cap and pre-warmed (39 °C) in an incubator. The incubation medium without rumen fluid was prepared the day before the study day using dipotassium phosphate (K_2_HPO_4_), 450; monopotassium phosphate (KH_2_PO_4_), 450; magnesium sulfate heptahydrate (MgSO_4_.7H_2_O), 190; calcium chloride dehydrate (CaCl_2_.2H_2_O), 120; Sodium chloride (NaCl), 900; cysteine hydrochloride (C_3_H_7_NO_2_S.HCl), 600; ammonium sulfate ((NH_4_)_2_SO_4_), 900; Trypticase peptone (BBL; Becton Dickinson, Cockeysville, MD, USA), 1000; and Yeast extract (Difco Laboratories, Detroit, MI, USA), 1000. The chemicals were poured in one liter of distilled water. Firstly, all the chemicals were poured and a very small amount of distilled water was added for the solution to mix evenly. Yeast extract and trypticase peptone were dissolved by hands since they clump immediately when these come in contact with air. Thereby, the immediate mixture of these chemicals was needed. In this process, a certain pH is required for the efficient function of the in vitro test and the required pH is 6.9 [20]. The pH was balanced by adding one to two drops of sodium hydroxide (NaOH) (Base) or hydrochloric acid (HCL). Afterwards, the buffer was dispensed with 100% nitrogen (N_2_) gas for creating anaerobic conditions. Lastly, the buffer was autoclaved at 121 °C for 15 min. Finally, the buffer was collected after almost one hour when the buffer was cooled after autoclaving and preserved until the next day for mixing with freshly rumen fluid. Rumen fluid was obtained before the morning feeding, mixed from both cannulated bulls at a 1:1 ratio, filtered through a double cheesecloth layer and 1000 mL was added to the buffer solution to prepare 4000 mL of total solution (rumen fluid: buffer solution; 1:3).

The 50 mL of the rumen fluid with the buffer solution was pumped into the 100 mL serum bottles, which contained 0.5 g sample of Pakchong grass of 6 LS, 9 LS and 14 LS with three SH of 5 cm, 10 cm and 20 cm. After that, the serum bottles were carefully handled to ensure absence of air bubbles on the serum bottle surface and placed into the incubator for 72 h at 39 °C. Five (5) experimental runs were conducted, where each run includes 9 samples (3 LS and 3 SH) with three replications per sample (total replications 15 each sample). Furthermore, each run had 27 serum bottles and their arrangement in the incubator was randomized. The cumulative gas production was recorded after 24, 48 and 72 h of incubation at 39 °C temperature for in vitro gas production with 120 rpm [21]. After 72 h, the feed samples were collected and washed through clean water and kept in the incubator at 60 °C for 48 h for determining the DM content [14]. A calibrated glass gas syringe made was used to collect the gas produced in the in vitro test. Fermentation parameters were monitored at the end of each incubation time set. A needle channel connected to the syringe was extended into the sealed fermentation bottle to measure the positive pressure created by the gas built up in the headspace of the syringe at room temperature and allowing the gas to flow inside a syringe barrel. The plunger was pulled gradually until the pressure the volume of gas trapped inside the barrel was recorded as the TG produced in ml.

Before in vitro rumen fermentation, the DM and organic matter (OM) of Napier feed sample was determined by drying at 60 °C for 48 h [14] and ashing at 550 °C for 7 h, respectively. The resulting DM and OM percentages were used to compute the initial DM (DM_i_) and initial OM (OM_i_) of the substrate in grams. Incubated samples from each serum bottle after the specified incubation period were drained in dried, pre-weighed nylon bags and knotted using nylon thread, then splashed with flowing water for 15 min or until the turbidity of water resulting from washing disappeared. The final DM (DM_f_) and OM (OM_f_) of the feed were determined using the same conditions applied when determining the initial values (DM_i_ and OM_i_). The DM and OM digestibility (%) were calculated as ([DM_i_ − DM_f_]/DM_i_) × 100 and ([OM_i_ − OM_f_]/OM_i_) × 100, respectively.

The ME was estimated by using the following equation: ME (MJ/kg DM) = 21.574–0.207 NDF [22].

### 2.5. Experiment 2: Cattle Growth

In this experiment, Napier grass was maintained and harvested to offer grass to cattle at 6, 9 and 14 LS, all harvested at 10 cm SH above ground level. Eighteen Red Chittagong cattle of 18–22 months of age growing bull, with an initial average live weight of 181.9 ± 2.30 kg (mean ± SE), were divided into three groups, namely 6 LS of Napier grass, 9 LS of Napier grass and 14 LS of grass. Each group included 6 cattle with an individual pen and offered with Napier grass harvested at 10 cm from the ground level at one of the three LS. All cattle were fed ad libitum and individually in a shed under stall fed conditions. The experiment was conducted for a period of 113 days, comprising a 7-day adjustment period, a 100-day feeding trial followed by a 7-day digestibility trial. No concentrate or supplementation was provided during the trial. Grass was supplied twice daily (at 8:00 a.m. and at 4:00 p.m.). Fresh and clean drinking water was supplied ad libitum to cattle throughout the experimental period. All cattle were weighed at an interval of 10 days. Growth of an individual was calculated from the final body weight deducted from the initial body weight, and the resultant was divided by the duration of feeding periods and finally expressed as g/day.

Feed intake of each cattle was monitored daily throughout the experiment. Daily feed intake was calculated from the differences in amount of offer and leftover that remained from that offer the next day. Before each feeding time in the morning and afternoon, grass to be offered was weighed and offered to the cattle in the individual feed trough. Leftovers were collected the next morning, weighed individually and recorded. The DM content of offered and remaining grass was measured every 10 days to align with the liveweight measurements. Fresh grass samples from all three LS treatments were collected twice a week for analysis (total sample: 26) and these samples were stored in a freezer at −18 °C. For proximate analysis, these collected samples were combined to produce one sample for each individual animal.

### 2.6. Feed Digestibility

For the digestibility trial, animals were kept in an individual metabolic crates and started 10 days prior to the conclusion of the experiment, with three days allocated for the adjustment to the new system and seven days for data collection. Leftover feed was weighed before the morning feeding, a sample was taken and stored at −18 °C for proximate analysis. Feces from each cattle were collected throughout the day (24 h) and kept in individual containers with lids. After 24 h, the total feces of each cattle were weighed, thoroughly mixed and a 5% subsample was collected in an empty container and stored in the freezer at −18 °C. Following the collection period, all stored fecal samples of each cattle were taken out of the freezer, thawed and carefully sorted, and around 300 g of mixed feces was taken for proximate analysis. Proximate analysis of feed, leftover and feces was analyzed following the methods mentioned above.

### 2.7. Season

In general, three types of cropping systems prevail in Bangladesh [23]: summer (Mid-March to mid-July), rainy (Mid-July to mid-November) and winter (Mid-November to mid-March). Therefore, the experimental period was divided into three seasons: summer, rainy and winter.

### 2.8. Statistical Analysis

In experiment 1, data on morphological characteristics, yield and nutritive value of Napier grass were subjected to analysis of variance (ANOVA) using a univariate GLM procedure based on a Randomized Complete Block Design (RCBD). The LS and SH were included as main effects. A least squares regression model in Statistical Package for the Social Sciences (SPSS) Version 20 computer software packages were used to describe statistical relationships between the treatment responses of a 3 × 3 factorial experiment with fixed effect of 3 LS at harvest (6, 9 and 14 LS), and 3 SH (applied as cutting/residual height) of 5, 10 and 20 cm, and the random effect of 3 blocks (B 1, 2, 3). Duncan test at 5% level of probability was applied as a post hoc test to compare the differences among treatment means. The statistical model was as follows:Y_ijkl_ = µ + LS_i_ + SH_j_ + B_k_ + LS_i_ × SH_j_ + E_ijkl_
where Y_ijk_ was the dependent variable, µ was population mean, LS_i_ = fixed effect of leaf stage (*i* = 1, 2, 3; i.e., 6, 9 and 14 leaf number), SH_j_ = fixed effect of severity height (*j* = 1, 2, 3; i.e., 5 cm, 10 cm and 20 cm), Bk = random effect of blocks *n* (*n* = 3), E_ijkl_ = residual error, assumed to be normally and independently distributed and LS_i_ × SH_j_ which were the fixed effect of ith leaf stage, jth severity height and their interaction, respectively.

Experiment 2, which was related to dietary treatments on intake, digestibility, nutritional quality and growth rate were analyzed statistically in an ANOVA of a completely randomized design (CRD) using GLM Procedures of SPSS Version 20 for Windows. Duncan’s Multiple Range test was used when the difference between treatment means was significant.

## 3. Results

### 3.1. Experiment 1

#### 3.1.1. Morphology

Increasing LS from 6 to 9 decreased leaf proportion (from 100 to 44%) but increased (*p* < 0.001) the stem proportion by more than 50% (from 0 to 56) (Table 2). In contrast, the leaf proportion increased (from 79–86%) but the stem proportion decreased significantly (from 21 to 14) as the SH increased from 5 to 20 cm (*p* < 0.001). They had the interaction effect on leaf and stem proportion, which was significant (*p* < 0.001).

Increasing the LS from 6 to 14 increased (*p* < 0.001) normal plant height (height at canopy or defoliation height) five times from 36 to 173 cm; a similar trend appeared for the raised plant height. Increasing LS from 6 to 14 also increased (*p* < 0.001) green leaves (from 6 to11), dead leaves (from 0 to 3) and number of nodes per plant (from 0 to 9). Defoliation interval (DI) was significantly shorter for 6 LS grass compared to those at 14 LS (mean 22 vs. 68 days). An opposite trend was noted for SH, increasing SH from 5 to 20 cm, reducing normal plant height from 74 to 68 cm, and raised plant height from 94 to 87 cm. Furthermore, raising SH from 5 to 20 resulted in a reduction in green leaves (from 8 to 7) and dead leaves (from 1 to 0.6) (*p* < 0.001), indicating a strong interaction effect. The days between harvests expanded from 22 to 68 days as LS grew from 6 to 14 (*p* < 0.001), resulting in a decrease in cutting number from 17 to 5. However, increasing SH from 5 to 20 resulted in a reduction in the days between harvest from 39 to 26 (*p* < 0.001), subsequently enhancing the cutting number from 9 to 14 (Table 2).

#### 3.1.2. Yield, Nutritive Value and In Vitro Digestibility

Raising LS from 6 to 14 significantly enhanced DM yield (*p* < 0.001) from 15.3 to 38.8 ton/ha/yr (Table 3). A comparable pattern was observed for SH; the DM yield rose (*p* < 0.01) from 24.1 to 27.1 ton/ha/yr as SH increased from 5 to 20 cm. An interaction impact was seen on year-round biomass yield (*p* < 0.05). Moreover, an increase in LS from 6 to 14 resulted in an enhanced CP yield from 2.8 to 4.4 ton/ha/yr (*p* < 0.001). However, SH had no impact on CP yield.

As LS increased from 6 to 14, DM, OM, ADF, NDF, ADL and silica content also increased (*p* < 0.001). In contrast, CP content decreased from 184 to 118 g/kg DM as LS increased, with similar CP across SH treatments. Similarly, ADF, NDF and ADL content decreased with increased SH. On the other hand, DM content increased as the SH increased (*p* < 0.001). However, increasing SH from 5 to 20, silica and OM increased from 28.6 to 34.6 g/kg DM (*p* < 0.01) and 918 to 931 g/kg DM (*p* < 0.05), respectively.

The dOM and ME of Napier grass were the highest (610 g/kg and 10.4 MJ/kg DM) at 6 LS 6 (*p* < 0.001) (Table 3). As the LS increased from 6 to 14, the dOM declined gradually from 610 to 520 g/kg, and ME decreased from 10.4 to 7.30 MJ/kg DM. However, SH had no effect on dOM and ME content of Pakchong grass.

#### 3.1.3. Minerals, Gross Energy, Nitrate-N, Water-Soluble CHO and Soluble Oxalate

Increasing LS from 6 to 14 decreased (*p* < 0.001) Ca (from 6.7 to 4.1 g/kg DM), K (from 13.8 to 7.8 g/kg DM), Mg (from 6.0 to 3.6 g/kg DM), P (from 3.3 to 1.8 g/kg DM), NO_3_-N (from 1.2 to 0.8 g/kg DM), WSC (from 153 to 123 g/kg DM) and soluble Ca-oxalate (from 34.8 to 18.0 g/kg DM) contents. However, LS did not affect the GE content of Napier grass (Table 4).

#### 3.1.4. Seasonal Impact on DM, CP Yield and Their Nutritional Values

Napier grass growth in all treatments began to decline in October, reaching its lowest production during December–February. Growth started to peak from March and peaked during July–August (Figure 1). The yearly average growth rate of Napier grass was 47, 66 and 109 kg DM/ha/day for 6, 9 and 14 LS, respectively. In summer (16 March to 30 June), the DM and CP yield increased gradually (1.43 to 8.56 ton/ha/cut) with an increase in LS from 6 to 14, (Figure 2) which was the highest production compared to winter (16 October to 15 March), and rainy (1 July to 15 October). However, the CP content of 6 LS was higher in the rainy season than 6 LS in the summer and winter seasons (Figure 3) and then decreased gradually from 9 LS to 14 LS. The rainy season produced the lowest amount of DM and CP, whereas the highest production was observed in the summer season) with different SH (Figure 4). Increasing SH from 5 to 20, the yield of DM and CP decreased gradually across the year. The CP content of 20 SH was the highest in the rainy season compared to 20 SH in summer and winter (Figure 5).

In addition, average DM and CP yield in the summer season was significantly (*p* < 0.001) higher (12.2 ton/ha and 1.60 ton/ha, respectively) followed by winter (6.34 ton/ha and 0.98 ton/ha, respectively) and rainy (4.62 ton/ha and 0.75 ton/ha, respectively). However, the CP content was significantly higher in the rainy season (167.5 g/kg DM) compared to the winter (156.2 g/kg DM) and summer (141.2 g/kg DM) (Table 5).

### 3.2. Experiment 2

#### 3.2.1. Nutrient Composition of Napier Grass Offered to Cattle

The nutrient composition (DM, OM, CP, ADF, NDF and ash) of Napier grass offered to cattle during the experiment differed significantly (*p* < 0.001) among treatments (Table 6). The crude protein content of the grasses was 155, 113 and 83.7 g/kg DM, and NDF content was 705, 718 and 737 g/kg DM, respectively, for 6, 9 and 14 LS of Napier grass treatments.

#### 3.2.2. Nutrient Intake and Digestibility of Napier Grass

Total dry matter (including % liveweight or W^0.75^ basis), ADF and NDF intake of cattle increased (*p* < 0.05), but CP intake decreased (*p* < 0.001) with the increase in LS from 6 to 14 (Table 7). Crude protein intake offered to cattle with 6 LS and 9 LS Napier grass was 1.8 and 1.3 times greater compared with 14 LS. Similarly, the DM, CP and NDF digestibility of grass in cattle offered 6 LS Napier grass was higher compared to that in cattle offered 9 or 14 LS Napier grass (Table 7).

### 3.3. Growth Performance and Feed Conversion Ratio (FCR)

Cattle on the 6 LS group had the greatest growth at 610 g/day, which was double that of the 14 LS Napier grass group (Table 8). The feed conversion ratio followed similar results with FCRs of 6 and 16 kg DM/kg LW gain for the 6 LS and 14 LS treatment.

## 4. Discussion

When Napier grass LS increased from 6 to 14 in experiment 1, there was a substantial decrease in CP from 184 to 118 g/kg DM, dOM (610 to 520 g/kg DM) and ME from 10.4 to 7.3 MJ ME/kg DM. Decreasing CP, dOM and ME with increasing LS of Napier grass in our experiment was associated with increases in NDF (from 540 to 691 g/kg DM), ADF (from 325 to 386 g/kg DM), ADL and silica contents (from 92.5 to 113 g/kg DM and 15.7 to 46.1 g/kg DM, respectively). These findings are consistent with other forages in the literature [1,2,3,11]. These changes in nutritional composition are attributed to a higher proportion of stem (increasing from 0 to 56%) and a greater number of dead leaves (from 0 to 3 dead leaves) as plant height (normal, not raised) increased from 36 to 173 cm and LS increased from 6 to 14. In addition, minerals, nitrate-N, WSC and soluble oxalate contents decreased as LS increased. These findings were consistent across seasons as seasonal variation had an impact on DM and CP contents at specific LS. The CP content of 6 LS was significantly higher in the winter season (Figure 4) compared to the other two seasons. However, SH did not have any impact on CP and DM content, which was supported by Tessema et al. [24]. As hypothesised, increase in fiber (and silica content), along with a decrease in CP, WSC and minerals, was associated with a reduced proportion of leaf [25] and an increased number of dead leaves as LS increased from 6 to 14. This pattern is likely to related to the observed decline in CP and ME content of Napier grass with greater LS. The aging process of the plant likely played a role in this decline as plant metabolism and nutrient mobilization shift with maturity [26,27]. Liman et al., 2020 [28] reviewed similar findings, showing that CP content decreases with plant aging. Nitrogen and WSC are mobilized from leaves for plant development [26,27], ultimately leading to increased fiber and reduced ME content of plants [29]. However, SH did not impact the CP and ME content of Napier grass in this experiment. In addition, the NDF, ADL and Silica content decreased gradually (*p* < 0.01) with the increase in SH from 5 to 20; however, no significant effect was found between 10 and 20 SH. These results emphasize the importance of a simple best management practice, such as the LS and SH management. For example, at 6 LS, Napier grass contained 184 g CP/kg DM and 10.4 MJ ME/kg DM compared with 118 g CP/kg DM and 7.27 MJ ME/kg DM at 14 LS with 10 cm SH from the ground level. Wijitphan et al. [30] research support this result and who showed that a 35-days interval and 15 cm from the ground level could be the best options for Napier grass. Machado et al. [31] reported similar findings, showing dOM to decrease from 75% to 55% as the interval between defoliations increased from 33 days to 93 days. Likewise, Goorahoo et al. [12] and Sileshi et al. [13] reported reductions in CP and ME content of Napier grass with increasing DI. These findings have significant implications for the livestock industry in the tropics and subtropics. Peyraud and Delagarde [32] reported that a 1% reduction in dOM of grass leads to a loss of 1 kg of milk/cow/day. Islam et al. [1] also reported that each per cent increase in CP in Napier grass corresponds to milk yield increase of 0.83 L/cow/day and each MJ increase in ME corresponds to a milk yield increase of 3.5 L/cow/day. Islam et al. [3] further highlighted the potential of Napier grass containing 200 g CP/kg DM, and 10 MJ ME/kg DM to produce >25,000 L milk/ha/yr, underscoring its role in achieving food security and potential to transform the lives of millions of people in the tropics and subtropics where Napier grass is a staple feed for dairy and beef cattle.

Despite the increase in Napier grass’s nutritive value as LS decreased, yield decreased by >50%. In line with our findings, Tessema et al. [24] reported that increasing DH from 100 to 300 cm doubled Napier grass yield, from 16 to 32 t DM/ha/yr. Similarly, Wangchuk et al. [33] also reported a significant increase in yield from 0.24 to 0.83 kg DM/plant when plant height increased from 150 to 260 cm. However, this increase in yield was accompanied by 81% greater biomass production resulting from the enhanced DH and a proportional reduction in the leaf proportion, and overall quality [12,13]. Thus, strategies are required to improve both quality and quantity simultaneously. Islam et al. [2] proposed a simple change, such as increasing plant density and reducing defoliation intervals, which is associated with LS in our experiment, to increase both yield and quality. Further research is needed to validate these strategies and explore their potential to optimize Napier grass production for livestock systems in the tropics and subtropics.

In experiment 2, the 6 LS group achieved a daily gain of 610 g by consuming only 6.44 kg DM, which contained 155 g CP/kg DM (Table 6). This growth rate was greater than the 210–250 g/day reported for young stock offered Napier grass containing <80 g of CP/kg of DM [34,35]. According to ARC [36] standard, grasses with 120 g CP/kg DM enable growing heifers to achieve a daily weight gain of 500 g, as they fulfill the rumen degradable protein requirements necessary for this growth rate. Additionally, a diet consisting solely of Napier grass at 56 days DI, containing 117 g CP/kg DM, resulted in a weight gain of 390–420 g LW/day, which was more effective than a Napier grass diet with 80 g CP/kg DM, yielding 210–250 g LW/day [34,35]. However, Holstein heifers consuming DM at a rate of 3.5% of LW and NDF at a rate of 2% of LW achieved 500 g/day LW. This was despite Napier grass, which was harvested at 42 days DI containing 8.6 MJ ME/kg DM, 118 g/kg DM CP and 587 g/kg DM NDF [37]. This growth of the heifers is consistent with the growth reported by ARC [36], indicating that cattle exclusively fed Napier grass containing approximately 120 g/kg DM CP and 550 g/kg DM NDF will grow at a rate of 500 g/day. The greater growth rate of cattle fed 50 cm DH Napier grass, despite the similar NDF content (550–570 g/kg DM) used in our experiment compared to those reported previously [36,37], is likely attributable to a difference in CP content. In our experiment, the CP content of grass was 180 g/kg DM compared to 120 g/kg DM reported in this study. Napier grass (Pakchong) in our experiment was harvested and offered to cattle in a rotation of 17-days interval (DI) compared to 42-days DI, reported by Kariuki et al. [37]. Interestingly, this Pakchong cultivar contained similar NDF at 17 days DI (57%) compared to that reported for 42 days DI (56%) by Kariuki et al. [37]. Several factors such as morphology, variety, inputs, maturity and DI influence the nutritive value of Napier grass [2]. Pakchong is considered a taller variety and likely to contain a higher proportion of stem compared to leafy varieties. Consequently, Pakchong may contain a higher NDF content compared to leafy cultivars, which may have been offered by Kariuki et al. [37]. In our experiment, cattle fed Napier grass containing 83.7 g/kg DM CP and 737 g/kg DM NDF (14 LS) grew at a rate of 270 g/day, which was approximately half of the gains reported by ARC [36] and Kariuki et al. [36]. Similarly, the LW gain of cattle was lower (350 g/day) when cattle were offered Napier grass with 113 g/kg DM CP and 718 g/kg DM NDF (9 LS) compared to previous reports [36,37]. These lower growth rates are likely to be due to high grass NDF content even when the CP levels of these grasses were relatively high.

Overall, our findings indicate that 6 LS is optimal under the conditions of our experiments without supplementing concentrates, but further work is required to evaluate this recommendation for other Napier grass cultivars. Managing Napier grass using this new best management practice may have significant implications for transforming animal production systems, thereby contributing to the food security of the millions of people in the tropics and subtropics.

## 5. Conclusions

Modifying the management procedures of Napier grass cv. Pakchong to harvest at 6 LS with a 10 cm SH markedly improved its nutritional quality, yielding 180 g/kg DM crude protein and 10 MJ ME/kg DM, resulting in improving animal production with a growth rate of 610 g/day, excluding concentrates. This presents a valuable opportunity to enhance food security for populations residing in the tropics and subtropics.

## Figures and Tables

**Figure 1 animals-15-01235-f001:**
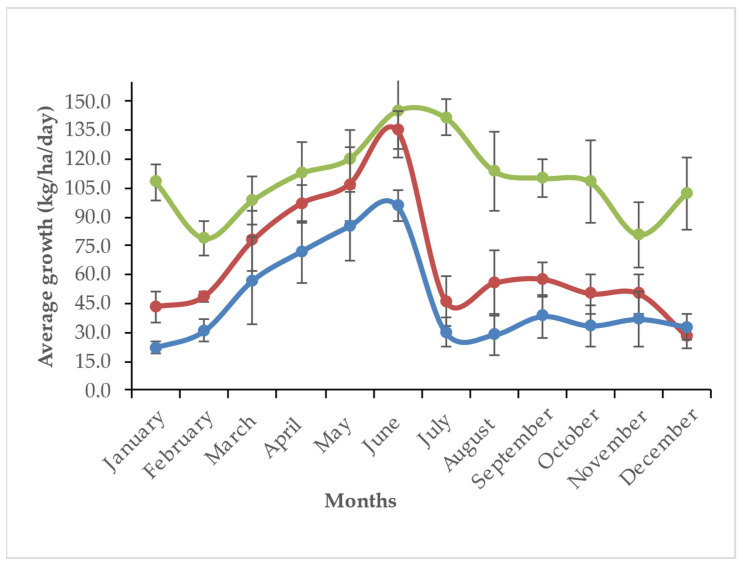
Growth rate of Napier grass at 6 (blue line), 9 (red line), and 14 (green line) leaf stage (LS) across the year.

**Figure 2 animals-15-01235-f002:**
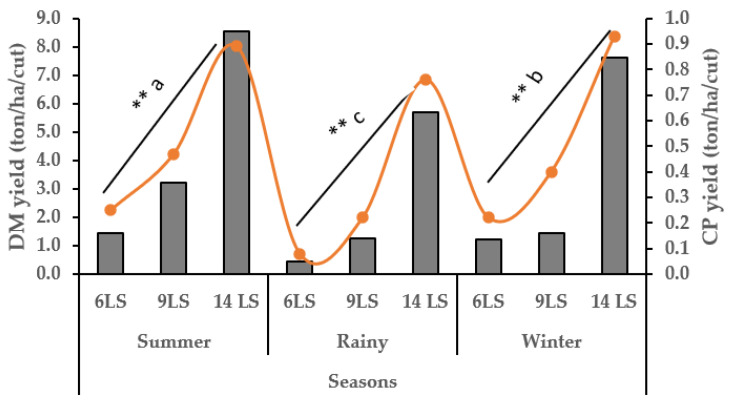
Seasonal impact on DM (grey columns) and CP (orange line) yield (ton/ha/cut) with different leaf stage (LS); Total number of cuts for 6 LS = 17; 9 LS = 11 and 14 LS = 5; ** = *p* < 0.01; ^a–c^ means with different superscripts in the same row are significantly different.

**Figure 3 animals-15-01235-f003:**
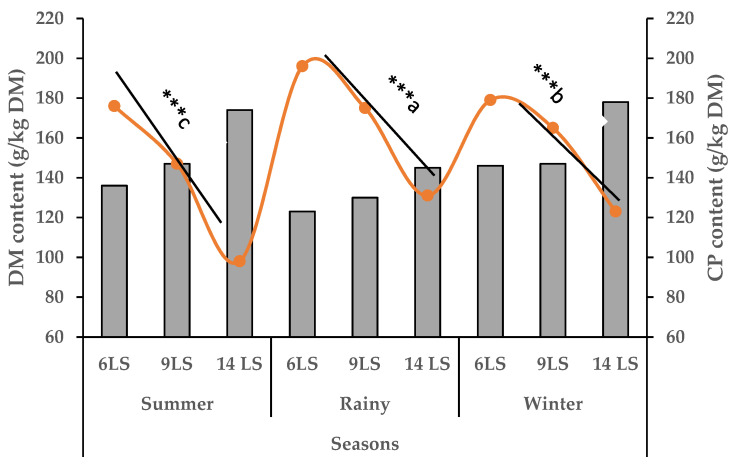
Seasonal impact on DM (grey columns) and CP (orange line) yield content (g/kg DM) with different leaf stage (LS); *** = *p* < 0.001; ^a–c^ means with different superscripts in the same row are significantly different.

**Figure 4 animals-15-01235-f004:**
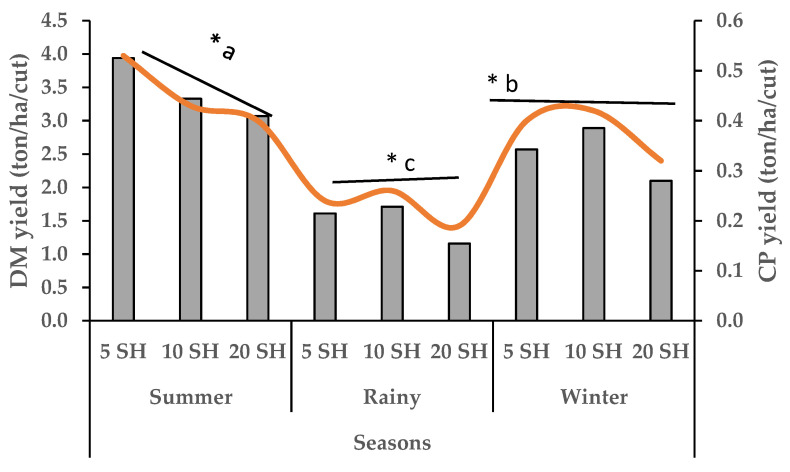
Seasonal impact on DM (grey columns) and CP (orange line) yield (ton/ha/cut) with different severity height (SH); Total number of cuts for 5 SH = 9; 10 SH = 10 and 20 SH = 14; * = *p* < 0.05; ^a–c^ means with different superscripts in the same row are significantly different.

**Figure 5 animals-15-01235-f005:**
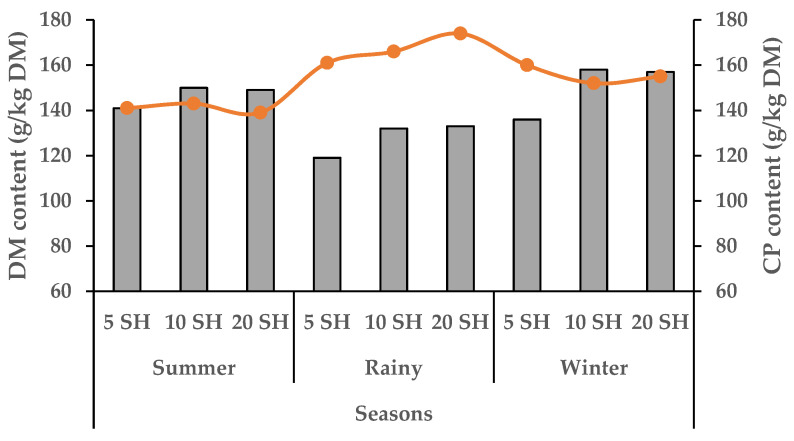
Seasonal impact on DM (grey columns) and CP (orange line) content (g/kg DM) with different severity height (SH).

**Table 1 animals-15-01235-t001:** Chemical composition of soil (dry soil).

Statistics	pH	OM (Organic Matter)	N (% Dry Matter)	K (ppm)	P (ppm)
Mean	5.3	1.71	0.09	0.17	6.32
SD	0.25	0.22	0.01	0.04	2.56

SD = standard deviation, OM = organic matter, N = nitrogen, K = potassium, P = phosphorus, ppm = parts per million.

**Table 2 animals-15-01235-t002:** Effect of leaf stage (LS) and severity height (SH) on morphological characteristics of Pakchong grass.

Parameters	6 LS			9 LS			14 LS			LS (No)			SH (cm)			SEM	Level of Sig.		
	SH			SH			SH			6	9	14	5	10	20		LS	SH	LS × SH
	5	10	20	5	10	20	5	10	20										
Leaf proportion	100	100	100	67	71	77	41	44	47	100 ^a^	72 ^b^	44 ^c^	79 ^b^	81 ^b^	86 ^a^	1.26	***	***	***
Stem proportion	0	0	0	33	29	23	59	56	53	0 ^c^	28 ^b^	56 ^a^	21 ^a^	19 ^b^	14 ^c^	1.26	***	***	***
Plant height (cm)																			
a. Normal	36	37	36	77	78	81	175	173	171	36 ^c^	79 ^b^	173 ^a^	74	73	68	2.82	***	NS	NS
b. Raised	52	52	52	101	101	101	200	200	199	52 ^c^	101 ^b^	199 ^a^	94	93	87	3.02	***	NS	NS
Leaf no./plant																			
a. Total	6	6	6	9	9	9	14	14	13	6 ^c^	9 ^b^	14 ^a^	9	9	8	0.19	***	NS	NS
b. Green	6	6	6	8	8	8	11	12	11	6 ^c^	8 ^b^	11 ^a^	8	8	7	0.14	***	NS	NS
c. Dead	0	0	0	1	1	1	3	2	2	0 ^c^	1 ^b^	3 ^a^	1 ^a^	0.9 ^a^	0.6 ^b^	0.06	***	***	***
Node no./plant	0	0	0	2.1	2.4	1.8	9.0	8.7	8.8	0 ^c^	2 ^b^	9 ^a^	2	2	2	0.19	***	NS	NS
Days between harvest	26	25	16	42	35	27	70	67	66	22 ^c^	33 ^b^	68 ^a^	39 ^a^	35 ^b^	26 ^c^	1.24	***	***	NS
Number of cuts	14	15	22	9	10	14	5	5	5	17	11	5	9	10	14	3.59			

LS, Leaf stage; SH, Severity height; SEM, Standard error of the mean; NS, Not significant (*p* > 0.05); *** = *p* < 0.001, ^a–c^ means with different superscripts in the same row are significantly different.

**Table 3 animals-15-01235-t003:** Effect of leaf stage (LS) and severity height (SH) on yield, nutritive value and in vitro digestibility of Pakchong grass.

Parameters	6 LS			9 LS			14 LS			LS (No)			SH (cm)			SEM	Level of Sig.		
	SH			SH			SH			6	9	14	5	10	20		LS	SH	LS × SH
	5	10	20	5	10	20	5	10	20										
Yield (ton/ha/yr)																			
DM	12.8	15.7	17.4	24.1	22.2	24.2	35.4	41.2	39.8	15.3 ^c^	23.5 ^b^	38.8 ^a^	24.1 ^b^	26.4 ^a^	27.1 ^a^	1.96	***	**	*
CP	2.4	2.8	3.2	3.7	3.6	3.9	4.4	4.5	4.2	2.8 ^c^	3.7 ^b^	4.4 ^a^	3.5	3.6	3.8	0.14	***	NS	NS
Chemical composition (g/kg DM)																			
DM	123	144	139	133	139	147	154	172	172	135 ^c^	140 ^b^	166 ^a^	136 ^b^	152 ^a^	153 ^a^	1.20	***	***	*
CP	185	182	185	159	163	167	118	117	118	184 ^a^	163 ^b^	118 ^c^	154	154	157	3.40	***	NS	NS
ADF	321	336	319	363	371	353	394	376	389	325 ^c^	362 ^b^	386 ^a^	359	360	354	5.37	***	NS	*
NDF	571	570	478	686	668	666	700	683	690	540 ^b^	673 ^a^	691 ^a^	653 ^a^	641 ^a b^	611 ^b^	15.3	***	*	NS
ADL	107	79.7	90.9	115	106	89	114	115	109	92.5 ^b^	103 ^a^	113 ^a^	112 ^a^	100 ^b^	97 ^b^	2.92	*	*	NS
Silica	21.2	11.0	14.9	17.4	35.2	37.1	47.5	39.0	51.9	15.7 ^c^	29.8 ^b^	46.1 ^a^	28.6 ^b^	28.4 ^b^	34.6 ^a^	2.82	***	**	***
OM	903	912	923	910	906	935	943	938	936	913 ^b^	917 ^b^	939 ^c^	918 ^b^	919 ^b^	931 ^a^	3.33	***	*	NS
In vitro digestibility																			
dOM (g/kg DM)	598	613	619	568	571	570	520	515	528	610 ^a^	570 ^b^	520 ^c^	562	566	572	7.17	***	NS	NS
ME (MJ/kg DM)	9.75	9.77	11.6	7.38	7.73	7.79	7.08	7.43	7.30	10.4 ^a^	7.63 ^b^	7.27 ^b^	8.07	8.31	8.92	0.32	***	NS	NS

LS, Leaf Stage; SH, Severity height; DM, Dry matter; CP, Crude protein; ADF, Acid detergent fiber; NDF, Neutral detergent fiber; dOM, Digestibility of organic matter; ME, Metabolizable energy; SEM, Standard error of the mean; NS, Not significant (*p* > 0.05); * = *p* < 0.05, ** = *p* < 0.01; *** = *p* < 0.001, ^a–c^ means with different superscripts in the same row are significantly different.

**Table 4 animals-15-01235-t004:** Effect of leaf stage on mineral content, gross energy (GE), nitrate–nitrogen (NO_3_-N), water-soluble carbohydrate (WSC) and soluble oxalate.

Parameters	Leaf Stage (No.)			SEM	Level of Sig.
	6	9	14		
Minerals (g/kg DM)					
Ca	6.7 ^a^	5.5 ^b^	4.1 ^c^	0.22	***
K	13.8 ^a^	13.8 ^a^	7.8 ^b^	0.60	***
Mg	6.0 ^b^	6.4 ^a^	3.6 ^c^	0.26	***
P	3.3 ^a^	2.2 ^b^	1.8 ^c^	0.22	***
Zn	0.07 ^b^	0.07 ^b^	0.09 ^a^	0.002	***
GE (MJ/kg, DM)	15.0	14.6	15.1	0.15	NS
NO_3_-N (g/kg DM)	1.2 ^a^	1.1 ^a^	0.8 ^b^	0.05	**
WSC (g/kg DM)	153 ^a^	129 ^a,b^	123 ^b^	9.20	*
Soluble Ca oxalate (g/kg DM)	34.8 ^a^	31.2 ^b^	18.0 ^c^	2.55	***

Ca, Calcium; K, Potassium; Mg, Magnesium; P, Phosphorus, Zn, Zinc; GE, Gross energy; NO_3_-N, Nitrate-N; WSC, Water soluble carbohydrate; SEM, Standard error of the mean; NS, Not significant (*p* > 0.05); * = *p* < 0.05, ** = *p* < 0.01; *** = *p* < 0.001, ^a–c^ means with different superscripts in the same row are significantly different.

**Table 5 animals-15-01235-t005:** Seasonal impact on yield and their nutritional value (DM and CP).

Parameters	Summer	Rainy	Winter	SEM	*p* Value
Yield (ton/ha)					
DM	12.2 ^a^	4.62 ^c^	6.34 ^b^	0.50	***
CP	1.60 ^a^	0.75 ^c^	0.98 ^b^	0.05	***
Chemical composition					
DM (g/Kg, fresh)	147.4 ^a^	128.6 ^b^	150.6 ^a^	0.08	***
CP (g/Kg DM)	141.2 ^c^	167.5 ^a^	156.2 ^b^	3.45	***

DM, Dry matter; CP, Crude protein; SEM, Standard error of the mean *** = *p* < 0.001, ^a–c^ means with different superscripts in the same row are significantly different.

**Table 6 animals-15-01235-t006:** Chemical composition of Napier grass offered to cattle.

Parameters	Leaf Stage (No.)			SEM	Level of Sig.
	6	9	14		
DM, Fresh basis (g/kg)	125 ^c^	145 ^b^	174 ^a^	2.27	***
OM (g/kg DM)	877 ^b^	878 ^b^	895 ^a^	1.12	***
CP (g/kg DM)	155 ^a^	113 ^b^	83.7 ^c^	2.23	***
ADF (g/kg DM)	406	412	410	1.76	NS
NDF (g/kg DM)	705 ^c^	718 ^b^	737 ^a^	2.36	***
Ash (g/kg DM)	123 ^a^	122 ^a^	105 ^b^	1.12	***

DM, Dry matter; OM, Organic matter; CP, Crude protein; ADF, Acid detergent fiber; NDF, Neutral detergent fiber; SEM, Standard error of the mean; NS, Not significant (*p* > 0.05); *** = *p* < 0.001, ^a–c^ means with different superscripts in the same row are significantly different.

**Table 7 animals-15-01235-t007:** Nutrient intake and digestibility of Napier grass.

Parameters	Leaf Stage (No.)			SEM	Level of Sig.
	6	9	14		
Nutrient intake (kg/day)					
DMI	3.92 ^b^	3.97 ^b^	4.30 ^a^	0.50	**
CPI	0.65 ^a^	0.47 ^b^	0.37 ^c^	0.10	***
ADFI	1.60 ^b^	1.65 ^b^	1.77 ^a^	0.02	**
NDFI	2.76 ^b^	2.85 ^b^	3.19 ^a^	0.04	***
DMI (Kg; % LW)	1.85 ^c^	1.99 ^b^	2.12 ^a^	0.02	***
DM intake (g, W^0.75^)	70.4 ^b^	74.7 ^a,b^	79.9 ^a^	0.88	*
Digestibility (g/kg DM)					
DM	630 ^a^	595 ^b^	567 ^b^	8.60	**
CP	684 ^a^	625 ^b^	592 ^b^	12.8	**
ADF	654	631	630	6.23	NS
NDF	697 ^a^	657 ^b^	637 ^b^	8.83	**

DMI, Dry matter intake; CPI, Crude protein intake; ADFI, Acid detergent fiber intake; NDFI, Neutral detergent fiber intake; LW, Live weight; SEM, Standard error of the mean; NS, Not significant (*p* > 0.05); * = *p* < 0.05, ** = *p* < 0.01; *** = *p* < 0.001, ^a–c^ means with different superscripts in the same row are significantly different.

**Table 8 animals-15-01235-t008:** Effect of sole Napier grass on growth performance and feed conversion ratio (FCR) of growing RCC cattle.

Parameters	Leaf Stage (No.)			SEM	Level of Sig.
	6	9	14		
Initial LW (Kg)	181	181	184	4.31	NS
Final LW (Kg)	246 ^a^	218 ^b^	213 ^b^	5.91	*
Total LW gain (Kg)	65.1 ^a^	37.4 ^b^	29.1 ^c^	3.98	***
ADG (Kg)	0.61 ^a^	0.35 ^b^	0.27 ^c^	0.04	***
FCR (kg DM/kg LW gain)	6.44 ^a^	11.3 ^b^	16.2 ^c^	1.05	***

LW, Live weight; ADG, Average daily gain; FCR, Feed conversion ratio; SEM, Standard error of the mean; NS, Not significant (*p* > 0.05); * = *p* < 0.05, *** = *p* < 0.001, ^a–c^ means with different superscripts in the same row are significantly different.

## Data Availability

None of the data were deposited in an official repository. The authors will provide the data used and analyzed during the current investigation upon request.

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
