# Peer review of "Management Strategies for Napier Grass (Pennisetum purpureum Schumach cv Pakchong): Impact on Dry Matter Yield, Nutritive Characteristics and Cattle Growth"

_animals, 2025, doi:10.3390/ani15091235_

Round 1

Reviewer 1 Report

Comments and Suggestions for Authors

General comments:

  1. Is the Napier grass used in this experiment from a specific strain or a variety developed through breeding? Significant trait and quality differences could be observed among different strains of Napier grass.
  2. The data currently presented in the analysis represents the average results from samples over the entire year. However, considering the substantial climatic differences among the different seasons in the cultivation area, the impact of seasonal environmental variations on Napier grass quality should be considered.
  3. The description of sample analysis and collection in the Materials and Methods section is overly brief, and further details on animal grouping and sampling procedures are required.
  4. The explanation of the SH factor in the results section is minimal, and the discussion lacks sufficient discussion of why different SH values did not affect DM and CP yields. If the collected grass samples include different harvesting SH under different LS conditions, there should be corresponding differences in the composition of the samples obtained. A simultaneous discussion of both factors (SH and LS) could provide insights into the optimal SH conditions for harvesting under various LS scenarios, rather than focusing solely on the selection of LS.
  5. If LS is the primary decision parameter for balancing yield and nutritional content, the conclusions of this experiment might be predicted before conducting the study. Proposing suitable SH values under various LS conditions or discussing seasonal variations in Napier grass growth and yield could offer more valuable insights.

Specific comments:

L72-75

Please clarify whether the referenced study specifically fed kikuyu grass alone. Was it fed as fresh grass or hay?

L150-151

Please provide the reference method of the drying conditions used.

L167-168

  1. The in vitro digestibility and gas production can be measured using the same equipment, but the Materials and Methods section should specify that the measurement was based on in vitro digestibility rather than gas production.
  2. What is the duration of the in vitro digestibility process? How to collect the sample of the digestion residual?

L173-174

Were the six cattle in each treatment group kept together or individually housed?

L175-177

Please provide the dates of the experiment. It will provide some information correlated with the growth rate of Napier grass, shown in Figure 1, and any potential environmental impacts.

L195-197 Why was the digestibility trial conducted only at the end of the experiment (after more than 100 days of feeding)?

L198-199

  1. How was the feces collected? If fecal bags were not used and the cattle were not housed individually, how could you ensure that each cattle's intake and excretion were recorded separately?
  2. How does the author prevent urine contamination during fecal collection?

L250

  1. Are there individual analysis data for various SHs under different LS conditions? If a 3x3 factorial design has been applied, individual data for all nine treatments should be provided before performing mean comparisons.
  2. If samples were collected monthly, how about the assay data about the composition changes of samples from different LS in each sampling month? Based on Figure 1, it appears that Napier grass growth varied significantly across months.

L256-257

How were dOM and ME calculated? This was not described in the Materials and Methods section. The energy determination using bomb calorimetry only provides gross energy values, so the basis for ME calculation should be described.

L279-280

  1. According to Figure 1, the growth rate of the 6 LS and 9 LS was relatively low from July to February of the following year. If Napier grass is harvested at the recommended stage (6 LS), can a single source of Napier grass provide sufficient biomass to meet the feeding needs? The stability of the feed supply and the seasonal variation in nutritional content should be discussed.
  2. If the CP, ADF, and NDF of Napier grass remain constant across seasons, what changes occur in other components? The composition assay data should be provided for clarification.

L284

Please remove the “caption for Figure 1.”

L323-325

The stems of Napier grass have a relatively high WSC content. When the LS is higher (plant aging), the proportion of stems increases, which could affect WSC and ADF content in the harvested sample.

L329-330

WSC% in Napier grass varies significantly throughout the day. The experiment should specify the exact time for switchgrass harvesting in the field, such as AM 8:00 or PM 3:00.

L336-337

Does SH affect the digestibility of CP and NDF? These data should be available from the in vitro trials. If LS is used as the primary parameter for harvest management, with a balance of nutritional content and yield, the experiment should demonstrate the actual dry matter yield and nutritional content under different LS x SH conditions.

L387-390

If “Pakchong” Napier grass is a variety with a higher proportion of stems, the digestibility of its NDF should be discussed.

L403-404

Since the experiment used only the “Pakchong” strain, the conclusion should not be generalized to suggest that all Napier grass should be harvested at 6 LS.

Author Response

For research article

Manuscript ID: animals-3520073

Summary

I sincerely appreciate your time spent reviewing this paper. Kindly refer to the comprehensive responses below together with the relevant amendments in the resubmitted documents.

Point-by-point response to Comments

Response to Reviewer 1 Comments

General Comments 1: [Is the Napier grass used in this experiment from a specific strain or a variety developed through breeding? Significant trait and quality differences could be observed among different strains of Napier grass.]

Response 1: statement agree. Literature said that Pakchong grass is a new hybrid grass crossing Pennisetum purpureum and Pennisetum glaucum known as Pearl Millet Napier.  It was first developed in Thailand in the late 1960s, and Department of Livestock Services (DLS) has been taken from Thailand to Bangladesh in 2015. After that Bangladesh Livestock Research Institute (BLRI) is doing their research with Pakchong grass. In 2020, BLRI performed a comparison study of 11 Napier cultivars conserved in the BLRI germplasm bank, revealing that Pakchong grass excelled in biomass and quality under existing management approaches. Consequently, we chose Pakchong Grass for its best management approaches, including LS and SH with feeding impact on growth performance of local bulls.

General Comments 2: [The data currently presented in the analysis represents the average results from samples over the entire year. However, considering the substantial climatic differences among the different seasons in the cultivation area, the impact of seasonal environmental variations on Napier grass quality should be considered.]

Response 2: We agree. We have updated the seasonal impact on Napier grass quality. The amendment is located on page 11 and 12 within the results section, notably lines 388-401 and 402, highlighted in red utilising the track change feature on the revised text.  

General Comments 3: [The description of sample analysis and collection in the Materials and Methods section is overly brief, and further details on animal grouping and sampling procedures are required]

Response 3: Thank you so much for thoroughly evaluating the manuscript. Based on our comments, we have described the details sample analysis and collection in the Materials and Methods section on pages from 173 to 203, 237-245, 262-267 and 275-278 highlighted in red utilising the track change feature on the revised text. 

General Comments 4: [The explanation of the SH factor in the results section is minimal, and the discussion lacks sufficient discussion of why different SH values did not affect DM and CP yields. If the collected grass samples include different harvesting SH under different LS conditions, there should be corresponding differences in the composition of the samples obtained. A simultaneous discussion of both factors (SH and LS) could provide insights into the optimal SH conditions for harvesting under various LS scenarios, rather than focusing solely on the selection of LS]

Response 4: We accept. We have described the SH in the results section on page 7, 8 and 9, which is included lines 318-325, and 352-367 highlighted in red utilising the track change feature on the revised text.  In the discussion section, we also described the LS and SH in the lines of 444-451 and 461-467 highlighted in red utilising the track change feature on the revised text. 

General Comments 5: [If LS is the primary decision parameter for balancing yield and nutritional content, the conclusions of this experiment might be predicted before conducting the study. Proposing suitable SH values under various LS conditions or discussing seasonal variations in Napier grass growth and yield could offer more valuable insights]

Response 5: We accept. Based on your comment, we have formulated the figure from 2-5 in the results section in the line of 402 and explained it line no 388-401, which will help to clear your conception.

Comments 5: [L72-75, Please clarify whether the referenced study specifically fed kikuyu grass alone. Was it fed as fresh grass or hay?]

Response 5: We appreciate this observation. Feed was offered as a fresh grass, which is mention line number 73 in Introduction section highlighted in red utilising the track change feature on the revised text. 

Comments 6: [L150-151 Please provide the reference method of the drying conditions used.]

Response 6: Your thorough review of the manuscript is greatly appreciated. Your observations are acknowledged and accepted. We have provided the reference method of drying conditions in the materials and methods section on the line 156-157 highlighted in red utilising the track change feature on the revised text.

Comments 7: [L167-168, 1. The in vitro digestibility and gas production can be measured using the same equipment, but the Materials and Methods section should specify that the measurement was based on in vitro digestibility rather than gas production.

  1. What is the duration of the in vitro digestibility process? How to collect the sample of the digestion residual?]

Response 7: We have explained the overall procedure of in vitro study with their duration in the materials and methods section in the lines from 173-230 highlighted in red utilising the track change feature on the revised text.

 Comments 8: [L173-174 Were the six cattle in each treatment group kept together or individually housed?]

Response 8: All animals have been kept in an individual pen mentioned in the materials and methods section specifically in the line of 240-242 highlighted in red utilising the track change feature on the revised text.

Comments 9: [L175-177 Please provide the dates of the experiment. It will provide some information correlated with the growth rate of Napier grass, shown in Figure 1, and any potential environmental impacts.]

Response 9: We have provided the experimental date in the materials and methods section, specifically in the line 94-95 highlighted in red utilising the track change feature on the revised text.

Comments 10: [L195-197 Why was the digestibility trial conducted only at the end of the experiment (after more than 100 days of feeding)?]

Response 10: In order to resolve this matter, it is possible to state that a digestibility trial was conducted in the metabolic crates, which was stressful and had an impact on the animal's growth. Consequently, we conducted this research at the end of the experiment.

Comments 11: [L198-199 How was the feces collected? If fecal bags were not used and the cattle were not housed individually, how could you ensure that each cattle's intake and excretion were recorded separately?

How does the author prevent urine contamination during fecal collection?]

Response 11: We have previously informed you that our digestible trial was conducted in metabolic crates individually, where a facility was available for the collection of urine and faces separately. Furthermore, we have employed additional labor to collect the dung immediately following its discharge, as there is no opportunity to mix it with urine.

Comments 11: [Are there individual analysis data for various SHs under different LS conditions? If a 3x3 factorial design has been applied, individual data for all nine treatments should be provided before performing mean comparisons.

If samples were collected monthly, how about the assay data about the composition changes of samples from different LS in each sampling month? Based on Figure 1, it appears that Napier grass growth varied significantly across months.]

Response 11: We have re-evaluated the data presented in Table 2 and Table 3, which encompass numerous SHs under distinct LS on pages 8 and 10 of the results section, namely lines 328-350 and 368-369, marked in red using the track change feature in the amended text.

We did not obtain the sample on a monthly basis; rather, we adhered to the LS with SH. This is the reason why it is not feasible to demonstrate the monthly compositional fluctuations. The seasonal impact on composition only DM and CP have been demonstrated in the results section of Figure 2 to Figure 5, specifically line numbers 402-403.

Comments 12: [L256-257 How were dOM and ME calculated? This was not described in the Materials and Methods section. The energy determination using bomb calorimetry only provides gross energy values, so the basis for ME calculation should be described.]

Response 12: We have added a detailed procedure on how to calculate dOM and ME in the Materials and Methods sections, specifically lines 228-230, highlighted in red, utilising the track change feature on the revised text.

Comments 13: [L279-280 According to Figure 1, the growth rate of the 6 LS and 9 LS was relatively low from July to February of the following year. If Napier grass is harvested at the recommended stage (6 LS), can a single source of Napier grass provide sufficient biomass to meet the feeding needs? The stability of the feed supply and the seasonal variation in nutritional content should be discussed.

If the CP, ADF, and NDF of Napier grass remain constant across seasons, what changes occur in other components? The composition assay data should be provided for clarification.]

Response 13: We are currently conducting additional research on plant density, which will result in a more than doubled biomass yield of 30-to-35 tons DM/ha/yr at 6 LS. This will assist in the recovery of the year-round green grass requirement. In addition, we are currently conducting an additional study on the production of silage using 6 LS of Napier grass, which will also address the feed scarcity.

We only measured the

 Based on your comment, we have added a detailed procedure on how to calculate dOM and ME in the Materials and Methods sections, specifically lines 228-230, highlighted in red, utilising the track change feature on the revised text.

Comments 14: [L284 Please remove the “caption for Figure 1.”]

Response 14: We have deleted the caption for Figure 1

Comments 15: [L323-325 The stems of Napier grass have a relatively high WSC content. When the LS is higher (plant aging), the proportion of stems increases, which could affect WSC and ADF content in the harvested sample.]

Response 15: Literature shows, with increasing LS, the ADF and WSC content decreased that I explained in the discussion section line no 441-443 and 444-447 highlighted in red, utilising the track change feature on the revised text.

Comments 16: [L329-330 WSC% in Napier grass varies significantly throughout the day. The experiment should specify the exact time for switchgrass harvesting in the field, such as AM 8:00 or PM 3:00.]

Response 16: We have collected samples from the research plot from 8:00 to 9:00 a.m., which is described in the Materials and Methods section, specifically in lines 149-152, highlighted in red, utilizing the track change feature on the revised text.  

Comments 17: [L336-337 Does SH affect the digestibility of CP and NDF? These data should be available from the in vitro trials. If LS is used as the primary parameter for harvest management, with a balance of nutritional content and yield, the experiment should demonstrate the actual dry matter yield and nutritional content under different LS x SH conditions.]

Response 17: There was no significant interaction between SH and LS for CP or NDF (Table 3).

Comments 18: [L387-390 If “Pakchong” Napier grass is a variety with a higher proportion of stems, the digestibility of its NDF should be discussed.]

Response 18: We have added line of 515-524 highlighted in red, utilising the track change feature on the revised text.  

Comments 19: [L403-404 Since the experiment used only the “Pakchong” strain, the conclusion should not be generalized to suggest that all Napier grass should be harvested at 6 LS.]

Response 19: We have included a statement line 549 focusing the recommendation on the Pakchong cultivar and recommending that further work be conducted to evaluate this recommendation for other Napier grass cultivars.

Response to Reviewer 2 Comments

Point-by-point response to Comments and Suggestions for Authors

Comments 1: [In material and methods, there was no information about soil type and reference to soil analysis. In addition, statistical analysis is incomplete.]

Response 1: We have described the materials and methods section in the line 115-116. We collected soil samples from different points and mixed them properly, made one sample, analyzed it, and performed the statistical analysis.  

Comments 2: [In the results, leaf proportion, stem proportion, dead leaves, DM, and ADF showed significance for interaction, but the authors did not write it in the text.]

Response 2: We have explained in the results section, specifically in the line of 315-318 highlighted in red, utilizing the track change feature on the revised text.   

Reviewer 2 Report

Comments and Suggestions for Authors

Dear All

I share my observation about the manuscript ID – animals – 3520073 Management strategies for Napier grass (Pennisetum purpureum Schumach): Impact on dry matter yield, nutritive characteristics, and cattle growth. The manuscript showed the results of two experiments. Experiment 1 studied the effects of the number of leaves and cutting height on the morphological characteristics and nutritional value of the Napier grass. Experiment 2 showed the effects of the number of leaves on cattle growth. In material and methods, there was no information about soil type and reference to soil analysis. In addition, statistical analysis is incomplete. In the results, leaf proportion, stem proportion, dead leaves, DM, and ADF showed significance for interaction, but the authors did not write it in the text. Moreover, only the effects of the number of leaves were analysed for the variables minerals, gross energy, nitrate-n, water-soluble CHO, and soluble oxalate.

Best regards!

Author Response

(The authors gave the same response as above.)

Round 2

Reviewer 1 Report

Comments and Suggestions for Authors

The content of the revised manuscript has significantly improved, and sufficient data has been provided to support the conclusions. Items listed below are the revision suggestions for the authors' further consideration.

General:

It is recommended that the outermost borders of the figures and tables be removed.

In the author’s response, some of the line numbers referenced do not correspond with the revised manuscript.

Specific:

L239-240
Please unify the superscripts and subscripts for T1-T3 and make corrections in conjunction with Table 8.

L237-238
Consider replacing the term "harvest number" here with "cutting number" as it may be easier to understand by Table 2.

Table 2
It is recommended to directly list the indicator data for all 9 treatment groups in the 3x3 design. The significance of differences between the 9 values should be indicated, and then the significance of the effects for LS, SH, and LS*SH can be displayed. This would help simplify the table.
The sample size (n) should be listed.

Fig 2 - Fig 5
The asterisks (*) and superscripts a, b, and c in the figures lack definitions in the legend.

Table 5
The title should be more precise. What does "their" refer to in the title?
The sample size (n) should be listed.

Author Response

For research article Manuscript ID: animals-3520073 Summary I sincerely appreciate your time spent reviewing this paper. Kindly refer to the comprehensive responses below together with the relevant amendments in the resubmitted documents. Point-by-point response to Comments Response to Reviewer 1 Comments General Comments 1: [It is recommended that the outermost borders of the figures and tables be removed.] Response 1: Thanks for your comment. Based on your comments, we have performed to remove the outermost borders of the figures and tables used in L331-L354, L379-L380, and L419. Comments 2: [L239-240 Please unify the superscripts and subscripts for T1-T3 and make corrections in conjunction with Table 8.] Response 2: Based on your comments, we have corrected specifically in the L239-240 and Table 8. Comments 3: [L237-238 Consider replacing the term "harvest number" here with "cutting number" as it may be easier to understand by Table 2.] Response 3: Again, thanks for thoroughly evaluating the manuscript. Based on your comments, we have replaced “harvest number” with cutting number in the Table 2. Comments 4: [Table 2 It is recommended to directly list the indicator data for all 9 treatment groups in the 3x3 design. The significance of differences between the 9 values should be indicated, and then the significance of the effects for LS, SH, and LS*SH can be displayed. This would help simplify the table. The sample size (n) should be listed.] Response 4: Agreed, which we showed before. However, it also explains the details of the experimental data. In addition, we have added the number of observations in Table 2 in blue color. Comments 5: [Fig 2 - Fig 5 The asterisks (*) and superscripts a, b, and c in the figures lack definitions in the legend.] Response 5: Based on your comment, we have performed in the line of L414. Comments 6: [Table 5 The title should be more precise. What does "their" refer to in the title? The sample size (n) should be listed.] Response 6: Thank you very much for your comments. Based on your comment, we have performed in Table 5 in the line of 417.